

# Effects of yearling, juvenile and adult survival on reef manta ray (*Manta alfredi*) demography

Isabel M. Smallegange[1], Isabelle B.C. van der Ouderaa[1] and Yara Tibiriçá[2]

[1] Institute for Biodiversity and Ecosystem Dynamics, University of Amsterdam, Amsterdam, Netherlands
[2] Zavora Marine Lab, Association of Coastal Conservation of Mozambique, Inharrime, Inhambane Province, Mozambique

## ABSTRACT

**Background**. The trade in manta ray gill plates has considerably increased over the last two decades. The resulting increases in ray mortality, in addition to mortality caused by by-catch, has caused many ray populations to decrease in size. The aim of this study was to ascertain how yearling and juvenile growth and survival, and adult survival and reproduction affect reef manta ray (*Manta alfredi*) population change, to increase our understanding of manta ray demography and thereby improve conservation research and measures for these fish.

**Methods**. We developed a population projection model for reef manta rays, and used published life history data on yearling and juvenile growth and adult reproduction to parameterise the model. Because little is known about reef manta ray yearling and juvenile survival, we conducted our analyses using a range of plausible survival rate values for yearlings, juveniles and adults.

**Results**. The model accurately captured observed variation in population growth rate, lifetime reproductive success and cohort generation time in different reef manta ray populations. Our demographic analyses revealed a range of population consequences in response to variation in demographic rates. For example, an increase in yearling or adult survival rates always elicited greater responses in population growth rate, lifetime reproductive success and cohort generation time than the same increase in juvenile survival rate. The population growth rate increased linearly, but lifetime reproductive success and cohort generation time increased at an accelerating rate with increasing yearling or adult survival rates. Hence, even a small increase in survival rate could increase lifetime reproductive success by one pup, and cohort generation time by several years. Elasticity analyses revealed that, depending on survival rate values of all life stages, the population growth rate is either most sensitive to changes in the rate with which juveniles survive but stay juveniles (i.e., do not mature into adults) or to changes in adult survival rate. However, when assessing these results against estimates on population growth and adult survival rates for populations off the coasts of Mozambique and Japan, we found that the population growth rate is predicted to be always most sensitive to changes in the adult survival rate.

**Discussion**. It is important to gain an in-depth understanding of reef manta ray life histories, particularly of yearling and adult survival rates, as these can influence reef manta ray population dynamics in a variety of ways. For declining populations in particular, it is crucial to know which life stage should be targeted for their conservation. For one such declining population off the coast of Mozambique, adult annual survival

Corresponding author
Isabel M. Smallegange,
i.smallegange@uva.nl

rate has the greatest effect on population growth, and by increasing adult survival by protecting adult aggregation sites, this population's decline could be halted or even reversed.

## INTRODUCTION

The global demand for animal products such as shark fins (*Clarke et al., 2006*), swim bladders (*Sadovy & Cheung, 2003*; *Clarke, 2004*), and ray gill plates (*White et al., 2006*; *Ward-Paige, Davis & Worm, 2013*) is unsustainable (*Berkes et al., 2006*; *Lenzen et al., 2012*). Since 1998, trading in products derived from manta and devil rays has increased exponentially (*Ward-Paige, Davis & Worm, 2013*). Ray gill plates are a key ingredient in traditional Chinese medicine, and cartilage serves as a filler in shark fin soup (*White et al., 2006*; *Ward-Paige, Davis & Worm, 2013*). The exploitation of ray species has resulted in population declines (*Marshall et al., 2011a*; *Couturier et al., 2012*), and increases their risk of extinction. As a result, some rays, including the reef manta ray *Manta alfredi* and giant manta ray *M. birostris*, are now listed as 'Vulnerable' on the International Union for Conservation of Nature (IUCN) Red List of Threatened Species (*Marshall et al., 2011a*). Reef manta rays have a life history strategy that results in late maturity, a long gestation period and a low mean lifetime reproductive success (*Marshall et al., 2011a*). Therefore, once a reef manta ray population starts to decrease or contains critically few individuals (e.g., due to overfishing), it is very difficult for the population to recover (*Kashiwagi, 2014*). Therefore, understanding how manta ray populations' growth rates are affected by variation in demographic rates such as growth, survival and fertility is particularly important (*Couturier et al., 2014*; *Kashiwagi, 2014*).

Recently, *M. alfredi* and *M. birostris* were listed on Appendix II of the Convention on International Trade in Endangered Species of Wild Fauna and Flora (CITES), meaning that any international trade in manta rays from September 2014 onward must be regulated. However, in many countries, particularly developing ones (e.g., Sri Lanka and countries in East Africa, such as Mozambique), manta ray populations are decreasing at an alarming rate (*Marshall et al., 2011a*; *Ward-Paige, Davis & Worm, 2013*). Although manta ray ecotourism occurs in many of these regions, only in 32% of them are manta rays protected (*Ward-Paige, Davis & Worm, 2013*). For example, despite their importance in ecotourism (*Tibiriçá et al., 2011*), manta rays are not protected under Mozambique law, despite the fact that there has been an 88% decrease in reef manta ray sightings off Praia do Tofo, Mozambique (*Rohner et al., 2013*). In addition, the main reef manta ray aggregation areas off the coast of southern Mozambique are not inside marine protected areas (*Pereira et al., 2014*), and there has been a rapid increase in the use of gill nets by artisanal fisheries within inshore regions that are frequented by the rays, which has significantly increased reef manta ray by-catch (*Marshall, Dudgeon & Bennett, 2011b*; *Pereira et al., 2014*). A comprehensive understanding of reef

manta ray demographics, and their responses to different mortality regimes, is therefore urgently needed to improve conservation efforts and management policies (*Ward-Paige, Davis & Worm, 2013*).

Although manta rays are often easy to approach, we currently do not have sufficient demographic data to fully understand their population dynamics (*Ward-Paige, Davis & Worm, 2013*). If conservation management policies are to be effective, knowing which age classes (yearlings, juveniles or adults) within a population are the most sensitive to disturbance is essential. For example, demographic analyses of the population dynamics of other long-lived organisms, such as turtles and killer whales (*Orcinus orca*), have revealed that population persistence is most sensitive to adult survival, whereas protecting young (e.g., through protective rearing schemes) has a much smaller impact on population persistence (*Brault & Caswell, 1993*; *Heppell, Crowder & Crouse, 1996*). Therefore, a very small decrease in the annual survival rate of adults likely has serious repercussions on the persistence of populations of long-lived species such as manta rays (*Ward-Paige, Davis & Worm, 2013*; *Kashiwagi, 2014*).

The aim of this study was to ascertain how yearling and juvenile growth and survival, and adult reproduction and survival, affect populations of reef manta rays. To this end, we developed a stage-structured population projection model (PPM) (*Caswell, 2001*) that we parameterised using published life history data obtained from populations off the coasts of southern Mozambique (*Marshall, Dudgeon & Bennett, 2011b*) and the Yaeyama Islands, Japan (*Kashiwagi, 2014*). Sufficient data were available to parameterise growth and reproduction in the PPM, but detailed information on the survival of yearling and juvenile reef manta rays is scarce (*Marshall et al., 2011a*; *Dulvy et al., 2014*); therefore, we used the model to investigate how different annual survival rates of yearlings, juveniles and adults affect the population growth rate, mean lifetime reproductive success and cohort generation time. We assessed the performance of the model by comparing the predicted values of these three population biology descriptors with empirical observations. Subsequently, we conducted elasticity analyses for all combinations of yearling, juvenile and adult survival rates to ascertain which demographic rate (rate at which individuals survive and stay in the same life stage; survive and grow into the next life stage; reproduce offspring) of which life stage (yearling, juvenile or adult) has the greatest influence on the population growth rate. Elasticity analysis is widely used by conservation biologists, because the results obtained can be used to prioritise conservation research and management for those life stages that have the greatest effect on population growth (*Benton & Grant, 1999*; *Carslake, Townley & Hodgson, 2009*). Because much less is known about yearling and juvenile survival rates than adult survival rates (*Marshall et al., 2011a*; *Dulvy et al., 2014*), investigating a range of yearling and juvenile survival rates will elucidate if and how reef manta ray population responses vary with variation in survival rates. For all of the combinations of yearling, juvenile and adult survival rates, we used the calculated population growth rates to project a population of 500 individuals forward over a period of 10 years, in order to investigate the population consequences of different yearling, juvenile and adult mortality regimes.
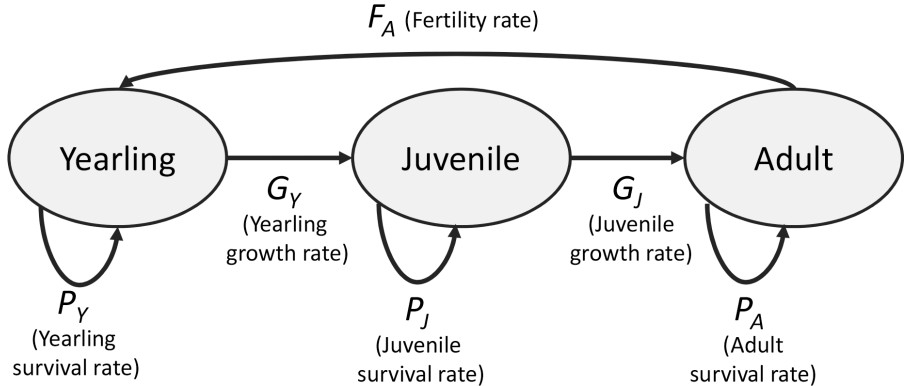

**Figure 1  Life cycle of *Manta alfredi*.** We distinguished three life stages: yearlings (Y), juveniles (J) and adults (A). The rate at which individuals survive and remain in the same life stage equals $P_i$, where $i$ indicates Y (yearling), J (juvenile) or A (adult); the rate at which individuals survive and grow to the next life stage equals $G_i$, where $i$ indicates Y (yearling) or J (juvenile); the rate at which adults produce yearlings equals $F_A$. See also Eqs. (1)–(3).

## METHODS

### *M. alfredi* life cycle

The life cycle of reef manta rays is generally divided into three life stages: yearlings, non-reproducing juveniles and reproducing adults (Fig. 1) (*Marshall et al., 2011a*; *Kashiwagi, 2014*). Male manta rays reach maturity after six years, and females are thought to mature at 8–10 years of age; longevity is estimated to be at least 40 years (*Marshall et al., 2011a*). On average, adult females produce one pup every two years, but fertility rates can range from one pup every one to five years (*Marshall et al., 2011a*). Reef manta ray life history data have been collected from various populations, including those off the coasts of Mozambique and the Yaeyama Islands, Japan (Table 1). These two populations differ remarkably in their estimated annual survival rates and population growth rates; the population off the coast of Japan is stable, and juveniles and adults exhibit high survival rates (0.95 per year) (*Kashiwagi, 2014*), whereas the population off the coast of Mozambique is declining, and the adult survival rate is estimated to be as low as 0.68 ± 0.147 SE (standard error) per year (*Marshall, Dudgeon & Bennett, 2011b*) (Table 1). In the present study, we used the life history data of these two populations to serve as reference points for our demographic analyses.

### Population model

The population model was based on a three-stage life cycle (Fig. 1). The addition of further life stages may have increased model accuracy, but these are the only currently distinguishable stages in *M. alfredi*. The rate at which individuals survive and remain in the same life stage (as opposed to growing into the next life stage) equals $P_i$ where $i$ indicates Y (yearling), J (juvenile) or A (adult), and was calculated following *Caswell (2001)*:

$$P_i = \sigma_i(1 - \gamma_i) \tag{1}$$

**Table 1  Life history data of different reef manta ray populations.** Shown are annual survival rates, $\sigma_i$, duration of different life stages, $D_i$, where $i = Y$ (yearlings), $i = J$ (juveniles) and $i = A$ (adults), and fertility rate of adults, $F_A$. Indicated are values estimated from data collected from populations off the coast of southern Mozambique and off the coast of Yaeyama Islands, Japan. Also shown are the values that were used in our demographic analyses.

|  | Explanation | Value in analyses | Observed value | Location of observation | Reference for observed value |
|---|---|---|---|---|---|
| $\sigma_Y$ | Annual survival rate of yearlings | 0.5–1.0 | 0.63 | Japan | *Kashiwagi (2014)* |
| $\sigma_J$ | Annual survival rate of juveniles | 0.5–1.0 | 0.95 | Japan | *Kashiwagi (2014)* |
| $\sigma_A$ | Annual survival rate of adults | {0.54, 0.68, 0.82, 0.95} | 0.68 | Mozambique | *Marshall, Dudgeon & Bennett (2011b)* |
|  |  |  | 0.95 | Japan | *Kashiwagi (2014)* |
| $D_Y$ | Duration of yearling stage (years) | 1 | 1 | Not specified/Japan | *Marshall et al. (2011a)*; *Kashiwagi (2014)* |
| $D_J$ | Average duration of (female) juvenile stage (years) | 9 | 8–10 | Not specified/Japan | *Marshall et al., 2011a*; *Kashiwagi (2014)* |
| $D_A$ | Duration of adult stage (years) | 31 | 31 | Not specified/Japan | *Marshall et al., 2011a*; *Kashiwagi (2014)* |
| $F_A$ | Average number of pups per year | 0.5 | 0.5 | Mozambique | *Marshall & Bennett (2010)* |

where $\sigma_i$ ($i = Y$, J, A) is the estimated survival rate for each life stage (Table 1). The parameter $\gamma_i$ is the transition rate from one life stage to the next (expressed per year); in this case, from yearling to juvenile ($\gamma_Y$) or from juvenile to adult ($\gamma_J$). Each transition rate $\gamma_i$ was calculated as ($\gamma_i = 1/D_i$) where $D_i$ is the duration (in years) of either the yearling ($i = Y$) or juvenile life stage ($i = J$) (Table 1). The rate (per year) at which individuals survive and grow into the next life stage is defined as:

$$G_i = \sigma_i \gamma_i \tag{2}$$

where $i$ indicates Y (yearling) or J (juvenile). The number of offspring produced each year equals $F_A$. These equations result in the following population projection matrix, which has a projection interval of one year:

$$\mathbf{A} = \begin{bmatrix} P_Y & 0 & F_A \\ G_Y & P_J & 0 \\ 0 & G_J & P_A \end{bmatrix}. \tag{3}$$

## Parameterisation and model performance

Following *Kashiwagi (2014)*, and as is common practice (*Caswell, 2001*), the population model was parameterised for females under the assumption that their growth and survival rates are not too dissimilar to those of male reef manta rays. We set the stage transition rates $\gamma_i$ in Eqs. (1) and (2) constant at $\gamma_Y = 1/D_Y = 1/1 = 1$ and $\gamma_J = 1/D_J = 1/8 = 0.125$ (Table 1), and assumed that females produce one pup every two years, so that $F_A = 0.5$ per year. Because little is known about yearling and juvenile survival rates (*Marshall et al., 2011a*; but see *Kashiwagi, 2014*), we conducted each demographic analysis (explained in the next section) for all combinations of values of yearling annual survival rate ($\sigma_Y$) and juvenile annual survival rate ($\sigma_J$) within the interval [0.5, 1] in increments of 0.005 (Table 1). We conducted each analysis using the observed adult annual survival rate of

reef manta rays off the coast of Mozambique, which is $\sigma_A = 0.68$ (*Marshall, Dudgeon & Bennett, 2011b*), a 20%-reduced adult annual survival rate of $\sigma_A = 0.54$, and 20%- and 40%-increased adult annual survival rates of $\sigma_A = 0.82$ and $\sigma_A = 0.95$, respectively (Table 1). The final value of $\sigma_A = 0.95$ is equal to the observed non-yearling annual survival rate of reef manta rays in a stable population off the coast of Japan (*Kashiwagi, 2014*) (Table 1). To assess the performance of our population model, we compared our predictions of population growth rate, lifetime reproductive success and cohort generation time with the empirical observations.

## Demographic analyses

Firstly, we calculated the population growth rate $\lambda$ from the dominant eigenvalue of matrix **A** (Eq. (3)) for each of the abovementioned combinations of yearling, juvenile and adult annual survival rates. Secondly, for each of the survival rate combinations, we performed an elasticity analysis to investigate how sensitive the population growth rate $\lambda$ is to perturbations of each of the different growth, survival and fertility rates in the population projection matrix **A** (Eq. (3)). To this end, we calculated the elasticity matrix **E**, where each element on row $m$ and column $n$ of matrix **E**, $e_{mn}$, represents the proportional contribution of each associated demographic rate $P_i$, $G_i$ and $F_A$ in the population projection matrix **A** (Eq. (3)) to the population growth rate $\lambda$. The elasticities were calculated as follows (*Caswell, 2001*):

$$e_{mn} = \frac{a_{mn}}{\lambda}\frac{\lambda}{a_{mn}} \tag{4}$$

where $a_{mn}$ are the elements of **A**. The second part of the equation describes the sensitivities of $\lambda$ to changes in the elements $a_{mn}$ of **A** (*Caswell, 2001*). The elasticities sum to 1, and give the proportional contributions of the matrix elements to the population growth rate $\lambda$. Therefore, the higher an elasticity value is relative to other elasticity values, the greater is the effect of the associated demographic rate on the population growth rate.

Thirdly, for each combination of yearling, juvenile and adult annual survival rates, we calculated mean lifetime reproductive success ($R_0$) by taking the dominant eigenvalue of the matrix **R** = **FN**. The matrix **F** is a fertility matrix that describes the production of new individuals:

$$\mathbf{F} = \begin{bmatrix} 0 & 0 & F_A \\ 0 & 0 & 0 \\ 0 & 0 & 0 \end{bmatrix}. \tag{5}$$

The matrix **N** is calculated as **N** = $(\mathbf{I} - \mathbf{U})^{-1}$, where **I** is the identity matrix and **U** the transient matrix that describes the growth and survival rates of the different stages:

$$\mathbf{U} = \begin{bmatrix} P_Y & 0 & 0 \\ G_Y & P_J & 0 \\ 0 & G_J & P_A \end{bmatrix}. \tag{6}$$

Fourthly, for each combination of yearling, juvenile and adult annual survival rates, we calculated cohort generation time as the mean age of offspring production in a cohort of

yearlings (*Caswell, 2009*):

$$T_c = \mathrm{diag}(\mathbf{FNe_Y})^{-1}\,\mathbf{FNUNe_Y} \tag{7}$$

where the vector $\mathbf{e}_Y$ is a vector with 1 in the first entry (for yearlings) and zeros in the second and third entries for juveniles and adults, respectively. Finally, we used the population growth rates that were calculated at step one to project a population of 500 individuals forward over a period of 10 years. All of the demographic analyses were conducted in MATLAB® R2014b (MathWorks®, MA, USA).

## RESULTS

### Model performance

Overall, the predictions from our PPM matched the empirical observations well. Firstly, the predicted values for the population growth rate $\lambda$ ranged from 0.64 to 1.13, depending on the values of yearling, juvenile and adult survival rates (Fig. 2 and Table 2). This range includes the range of observed population growth rate values, but also slightly exceeds the range of observed values (Table 2). Similarly, the range of predicted values of lifetime reproductive success $R_0$ (0.06–6.20) (Fig. 3 and Table 2) included the range of observed values of $R_0$, but the highest predicted value of $R_0$ exceeded the highest observed value of $R_0$ (Table 2). The predicted values for cohort generation time were very low (Fig. 4 and Table 2), and much lower than the observed cohort generation times in most cases (Table 2). Only when adult annual survival rate was at its highest ($\sigma_A = 0.95$) (Fig. 4D) did the predicted cohort generation time match the observed value (Table 2).

### Summary of the demographic analyses

Because the results of our demographic analyses are complex, we first provide a summary to aid in the interpretation of the specific results (below). Because little is known about survival rates of yearling and juvenile reef manta rays, we explored the effects of a range of values of yearling and juvenile annual survival rates on lifetime reproductive success, population growth rate and cohort generation time. We also varied the adult annual survival rate from as low as 0.54, which is 20% lower than the observed annual survival rate of adults (0.68 per year) off the coast of Mozambique (*Marshall, Dudgeon & Bennett, 2011b*), to as high as 0.95 per year, which equals the observed adult annual survival rate in the stable population off the coast of the Yaeyama Islands (*Kashiwagi, 2014*). The effect of an increase in adult annual survival rate across this range of values was straightforward: with increasing adult annual survival rate, the values of all three population descriptors also increased. However, variation in yearling and juvenile annual survival rates had different and varying effects on the population descriptors that we investigated. In the case of population growth rate, changes in these two survival rates had additive effects on the population growth rate, but interactive (multiplicative) effects on mean lifetime reproductive success, whereas cohort generation time was unaffected by variation in the juvenile annual survival rate. In addition, the effect of an increase in juvenile annual survival rate was always of a far greater magnitude on population growth rate, mean lifetime reproductive success and cohort

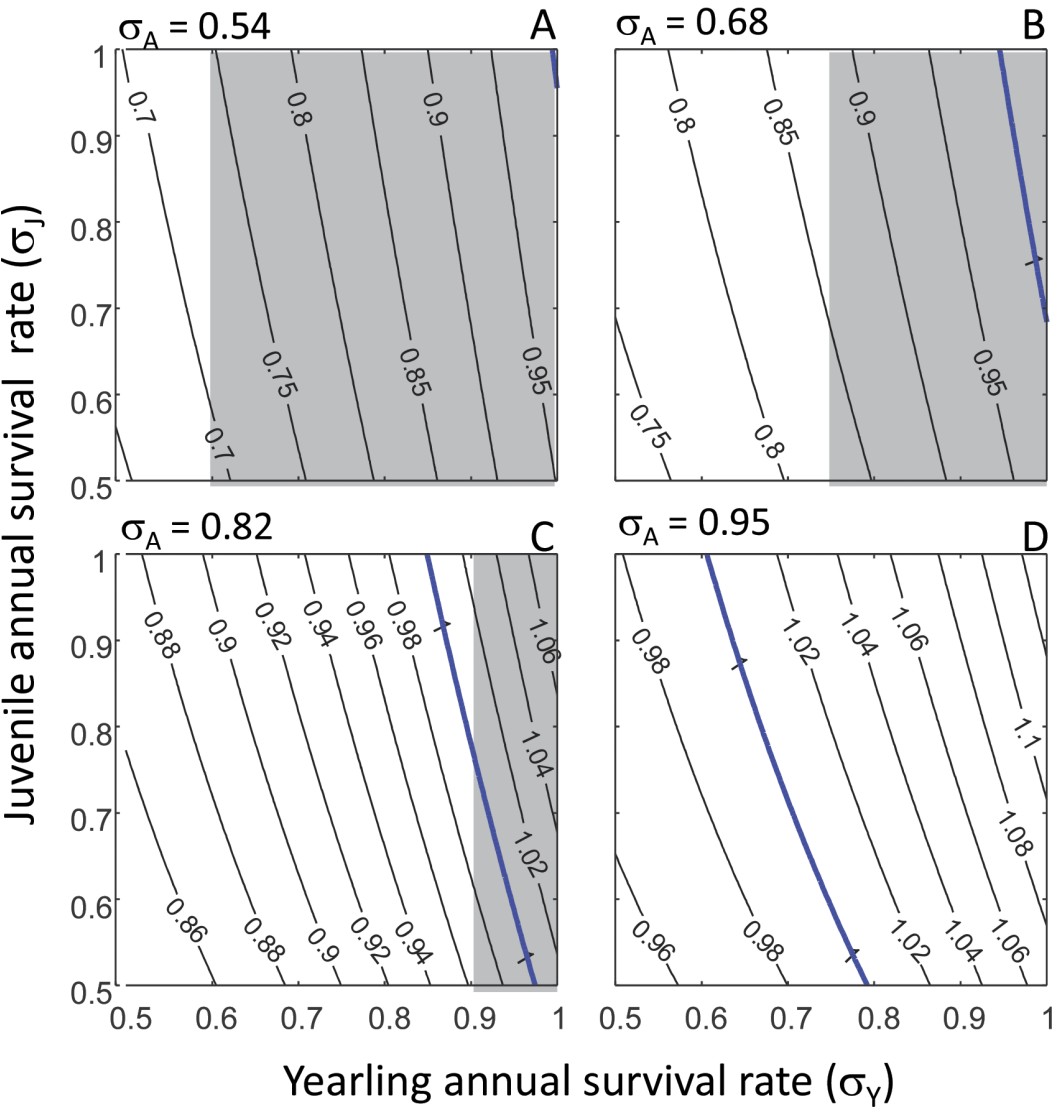

# Population growth rate (λ)

**Figure 2  Population growth rate and elasticity results.** Predicted population growth rate λ in relation to yearling annual survival rate ($\sigma_Y$) and juvenile annual survival rate ($\sigma_J$) shown for each of four values of adult annual survival rate ($\sigma_A$): $\sigma_A = 0.54$ (80% of observed rate) (A); $\sigma_A = 0.68$ (observed rate) (B); $\sigma_A = 0.82$ (120% of observed rate) (C); and $\sigma_A = 0.95$ (140% of observed rate) (D). In each panel, isoclines denote equal values of the population growth rate λ. The blue line in each panel denotes population stability at λ = 1; values higher than λ = 1 denote increasing populations and value lower than λ = 1 denote declining populations. The grey and white areas in panels denote the elasticity results: white areas (panel D is all white) denote parameter combinations where the population growth rate is most sensitive to $P_A$, the rate at which adults survive and remain in the adult stage (Eq. (3)); grey areas denote parameter combinations where the population growth rate is most sensitive to $P_J$, the rate at which juveniles survive and remain in the juvenile life stage (Eq. (3)).

![PeerJ]

**Table 2 Predicted and observed population descriptors for different reef manta ray populations.** The population descriptors are: population growth rate ($\lambda$, expressed per year), mean lifetime reproductive success ($R_0$), and cohort generation time ($T_c$, years). Predicted values given are the minimum and maximum values from our demographic analyses (Figs. 2–4); observed values are taken from different locations around the world (locations are indicated).

| | Predicted range | Observed value | Explanation of observed value | Location of observation | Reference for observed value |
|---|---|---|---|---|---|
| $\lambda$ | 0.64–1.13 | 0.77 | Calculated from the observation of 88% decline between 2005–2011 | Mozambique | *Rohner et al. (2013)* |
| | | 0.98 | Calculated from the observation of 80% decline over 75 years | Not specified | *Marshall et al. (2011a)* |
| | | 1.02 | Estimated using POPAN models covering 1987–2009 | Japan | *Kashiwagi (2014)* |
| $R_0$[a] | 0.06–6.20 | 0.60 | Calculated using IUCN data (*Marshall et al., 2011a*)[a]: $T_c = 25$ and $\lambda = 0.98$ | Not specified | *Marshall et al. (2011a)* |
| | | 0.02 | Worst-case scenario calculated using slowest life history values[a]: $T_c = 19.4$ and $\lambda = 0.77$ | Not specified | *Marshall et al. (2011a)* and *Rohner et al. (2013)* |
| $T_c$ | 3.89–20.40 | 19.4 | Mean of minimum (6.75 years) and maximum (32 years) age of adults | Tropical Eastern Pacific & Atlantic; Hawaii | *Ward-Paige, Davis & Worm (2013)* |
| | | 25 | Mean of minimum (10 years) and maximum (40 years) age of adults | Not specified | *Marshall et al. (2011a)* |

**Notes.**
[a]$R_0$ was calculated by taking the exponent of $T_c \times \log(\lambda)$ (*Caswell, 2001*).

generation time than the effect that the same increase in yearling or adult annual survival rate had on these population descriptors.

## Specific results of the demographic analyses

Calculating the population growth rate $\lambda$ for all of the different values of yearling, juvenile and adult annual survival rates revealed that for the observed adult annual survival rate of $\sigma_A = 0.68$ (*Marshall, Dudgeon & Bennett, 2011b*), populations can only persist if both yearling and juvenile annual survival rates are high ($\sigma_Y > 0.7$ and $\sigma_J > 0.95$) (Fig. 2B; populations persist to the right of the blue line, indicating population stability at $\lambda = 1$). When applying the lower value of adult annual survival rate ($\sigma_A = 0.54$), populations can only persist if both yearling and juvenile annual survival rates are almost at unity (Fig. 2A; populations persist to the right of the blue line, indicating population stability at $\lambda = 1$). At higher values of $\sigma_A$ ($\sigma_A = 0.82$ and $\sigma_A = 0.95$), populations can persist at much lower values of yearling and juvenile annual survival rates (Figs. 2C and 2D; populations persist to the right of the blue line, indicating population stability at $\lambda = 1$), e.g., if $\sigma_A = 0.95$, the yearling survival rate ($\sigma_Y$) can be as low as 0.5, as long as the juvenile survival rate is $\sigma_J = 0.8$ (Fig. 2D). Because the isoclines in each panel are neither horizontal nor vertical, we can infer that for a constant value of $\sigma_Y$ (or $\sigma_J$), the population growth rate depends on what the value of $\sigma_J$ (or $\sigma_Y$) is. However, because the isoclines in each plot are parallel, we can infer that these effects are additive, and that therefore there is no interactive effect between $\sigma_Y$ and $\sigma_J$ on $\lambda$ (i.e., the magnitude of an effect of $\sigma_Y$ on $\lambda$ does not depend on the value of $\sigma_J$, and vice versa).

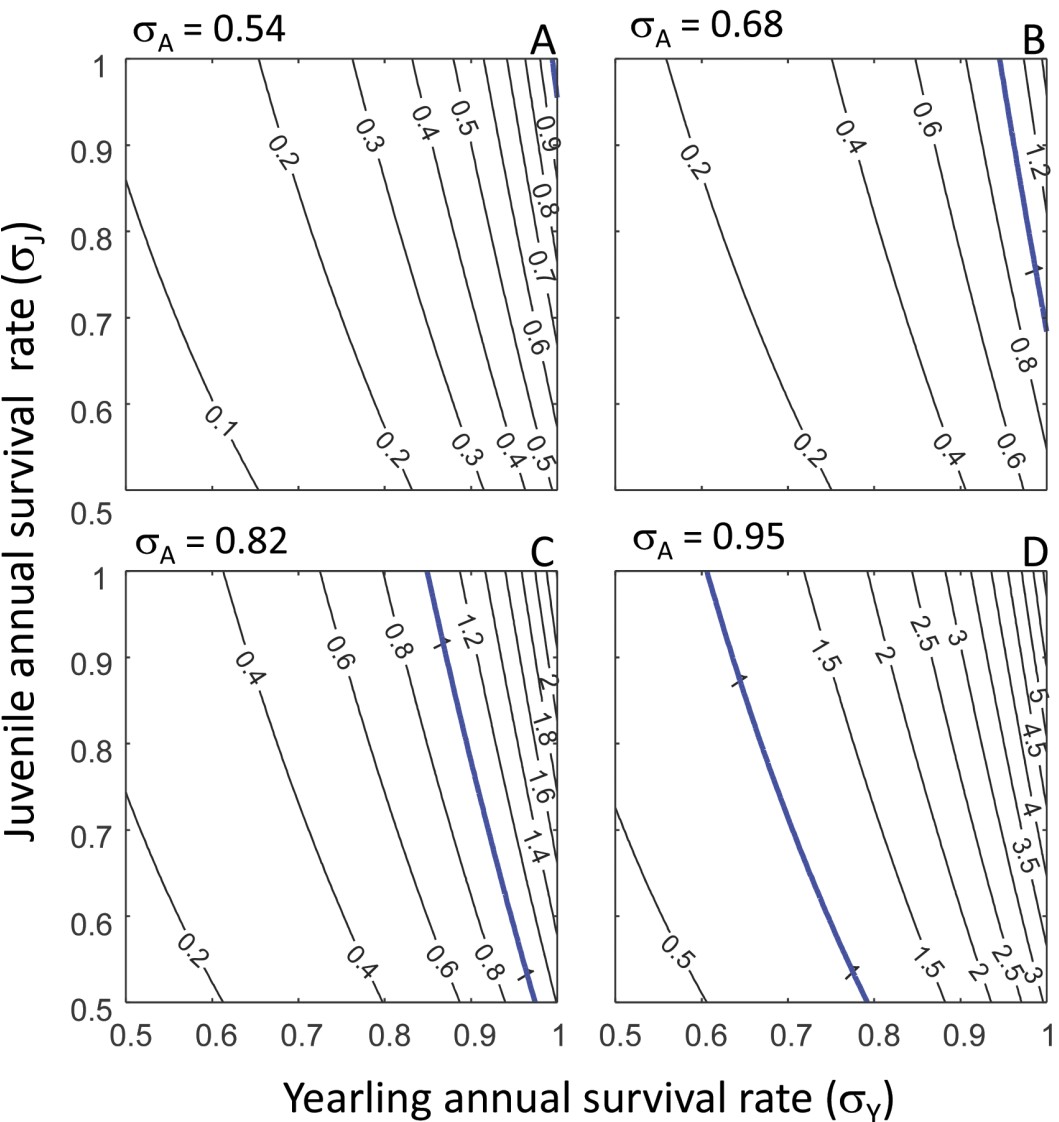

**Figure 3** **Mean lifetime reproductive success.** Predicted lifetime reproductive success ($R_0$) in relation to yearling annual survival rate ($\sigma_Y$) and juvenile annual survival rate ($\sigma_J$) shown for each of four values of adult annual survival rate ($\sigma_A$): $\sigma_A = 0.54$ (80% of observed rate) (A); $\sigma_A = 0.68$ (observed rate) (B); $\sigma_A = 0.82$ (120% of observed rate) (C); and $\sigma_A = 0.95$ (140% of observed rate) (D). In each panel, iso-clines denote equal values of lifetime reproductive success, $R_0$. The blue line in each panel denotes population stability at $R_0 = 1$; values higher than $R_0 = 1$ denote increasing populations and value lower than $R_0 = 1$ denote declining populations.

We then investigated how variation in yearling, juvenile and adult survival rates affected the elasticity of the population growth rate $\lambda$ to each of the demographic rates of each life stage in the PPM (Eq. (3)). This revealed that, depending on the survival rate values, $\lambda$ was either most sensitive to $P_A$, the rate at which adults survive and remain in the adult stage, or $P_J$, the rate at which juveniles survive and remain in the juvenile life stage (Fig. 2;

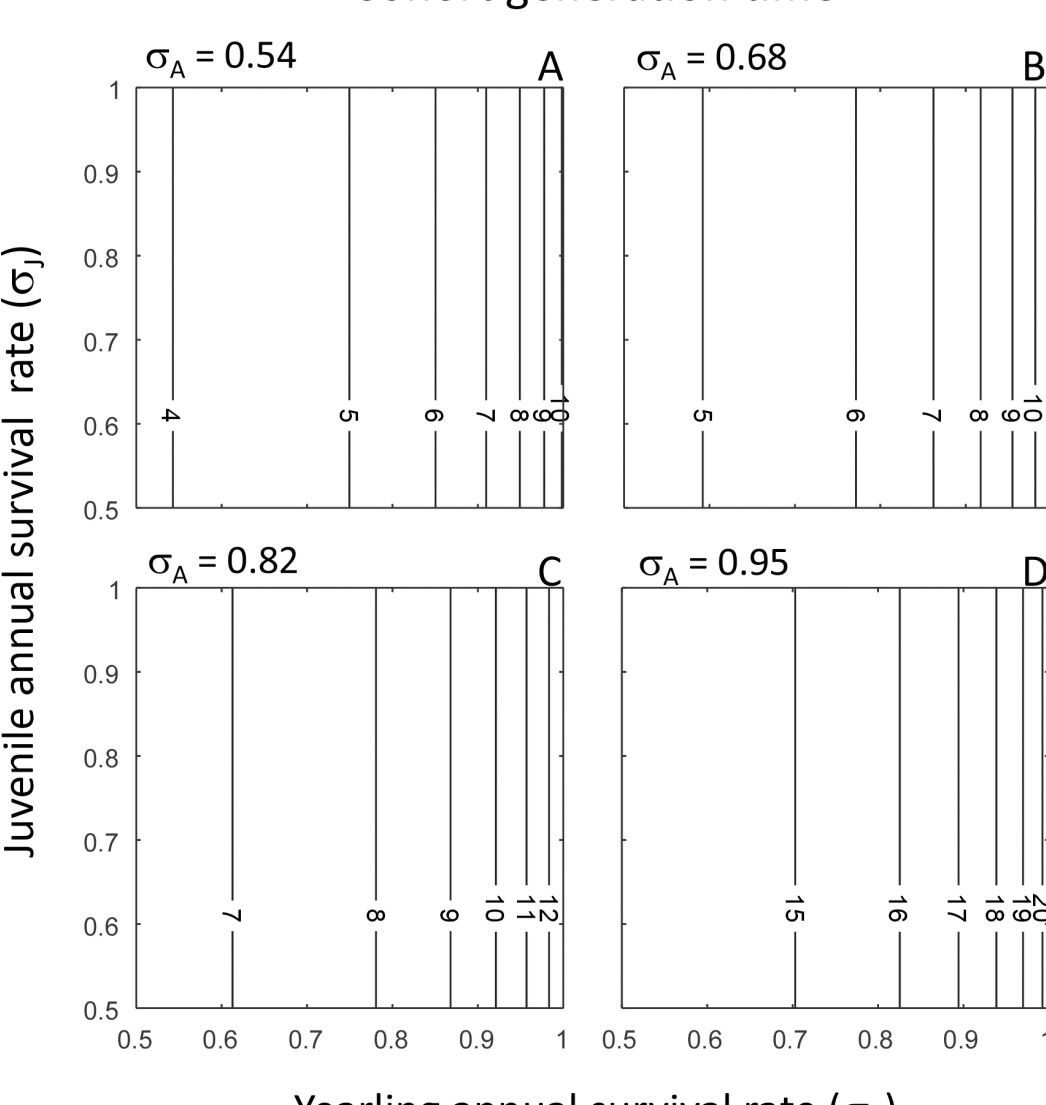

**Figure 4  Cohort generation time.** Predicted cohort generation time ($T_c$) in relation to yearling annual survival rate ($\sigma_Y$) and juvenile annual survival rate ($\sigma_J$) shown for each of four values of adult annual survival rate ($\sigma_A$): $\sigma_A = 0.54$ (80% of observed rate) (A); $\sigma_A = 0.68$ (observed rate) (B); $\sigma_A = 0.82$ (120% of observed rate) (C); and $\sigma_A = 0.95$ (140% of observed rate) (D). In each panel, isoclines denote equal values of cohort generation time.

white areas in each panel denote survival rate values where $\lambda$ is most sensitive to $P_A$, and grey areas denote survival rate values for which $\lambda$ is most sensitive to $P_J$). Interestingly, with increasing values of the adult annual survival rate $\sigma_A$ (going from Figs. 2A to 2D), the region of yearling survival rate ($\sigma_Y$) values for which $\lambda$ is most sensitive to $P_A$ (white areas) increases, whereas the region of yearling survival rate values for which $\lambda$ is most sensitive to $P_J$ (grey areas) decreases. These shifts indicate that the elasticity results were

independent of juvenile annual survival rate ($\sigma_J$); rather, whether or not $\lambda$ is most sensitive to perturbations in $P_J$ or $P_A$ critically depended on the values of $\sigma_Y$ and $\sigma_A$ (Fig. 2).

We then investigated the effect of variation in yearling, juvenile and adult survival rates on mean lifetime reproductive success. The results were qualitatively similar to those obtained for the population growth rate: with increasing values of the adult annual survival rate $\sigma_A$, populations can persist at ever lower values of yearling and juvenile annual survival rates (Fig. 3; populations persist to the right of the blue line, indicating population stability at $R_0 = 1$). In contrast to the results for population growth rate, however, the isoclines in each panel are not parallel and are unevenly spaced (Fig. 3), indicating that the yearling and juvenile annual survival rates $\sigma_Y$ and $\sigma_J$ have an interactive effect on lifetime reproductive success, i.e., the magnitude of an effect of $\sigma_Y$ on lifetime reproductive success depends on the value of $\sigma_J$, and vice versa. The uneven spacing of the isoclines for each value of adult annual survival rate (Fig. 3) indicates that, with increasing values of yearling and juvenile annual survival rates, lifetime reproductive success increases at an ever higher rate. This increase in lifetime reproductive success is greater with increasing values of yearling or adult annual survival rates than with increasing values of juvenile annual survival rates.

Regarding cohort generation time, for each value of adult annual survival rate ($\sigma_A$), cohort generation time increases at an accelerating rate with increasing values of yearling annual survival rate ($\sigma_Y$); hence, a slight increase in a high value of $\sigma_Y$ results in a far larger increase in cohort generation time than a slight increase in a low value of $\sigma_Y$. However, there is no effect of juvenile annual survival rate ($\sigma_J$), because the increase in cohort generation time with increasing values of $\sigma_Y$ is the same for each value of $\sigma_J$ (Fig. 4). Cohort generation time also increases at an accelerating rate with increasing values of adult annual survival rate ($\sigma_A$); consequently, a slight increase in a high value of $\sigma_A$ results in a far larger increase in cohort generation time than a slight increase in a low value of $\sigma_A$ (Fig. 4).

Finally, we used the predicted population growth rates (Fig. 2) to project a starting population of 500 individuals forward over 10 years to investigate the population consequences of variation in yearling, juvenile and adult survival rates. The combinations of yearling, juvenile and adult survival rate values at which populations are stable and the projected population size remains at 500 individuals after 10 years (indicated by the green lines in Fig. 5) were the same as those obtained from the population growth rate (Fig. 2) and lifetime reproductive success (Fig. 3) analyses. In each panel in Fig. 5, combinations of survival rate values to the right of the green line correspond to population increases. Comparing the different panels shows that the increase in population size is greater at higher values of the adult annual survival rate ($\sigma_A$) (Fig. 5). The lowest observed population size of reef manta rays off the coast of Mozambique is 149 individuals (*Marshall, Dudgeon & Bennett, 2011b*), and is indicated by red lines in Fig. 5. Matching this lowest observed population size to our population projections reveals that it corresponds to ever lower values of yearling annual survival rate ($\sigma_Y$) and juvenile annual survival rate ($\sigma_J$) as the adult annual survival rate ($\sigma_A$) increases in value. This suggests that the decrease in population size over 10 years is less at higher values of the adult annual survival rate than at lower values.

**Figure 5  Population size projected over ten years.** A population of 500 individuals is projected over ten years using the predicted population growth rate λ (Fig. 2). Projected population sizes are shown in relation to yearling annual survival rate ($\sigma_Y$) and juvenile annual survival rate ($\sigma_J$) for each of four values of adult annual survival rate ($\sigma_A$): $\sigma_A = 0.54$ (80% of observed rate) (A); $\sigma_A = 0.68$ (observed rate) (B); $\sigma_A = 0.82$ (120% of observed rate) (C); and $\sigma_A = 0.95$ (140% of observed rate) (D). In each panel, isoclines denote equal values of projected population size. The green line in each panel denotes population stability where the projected population size is equal to the initial size of 500 individuals; above and below this line, populations are projected to increase or decrease respectively. The red line in each panel depicts a population size of 149 individuals, which is equal to the lowest observed population size of reef manta rays off the coast of Mozambique (*Marshall, Dudgeon & Bennett, 2011b*).

## DISCUSSION

### Model performance

In this study, we developed a population model for reef manta rays that we used to conduct a detailed analysis of reef manta ray demography. With this analysis, we aim to increase our understanding of the drivers of population change in reef manta rays, and how perturbations to demographic rates, such as a decrease in survival due to targeted fishing and by-catch, affect their population fluctuations. Before we discuss our findings against current understanding of reef manta ray demography, however, it is necessary: (i) to know how well our PPM performed in describing the general characteristics of reef manta ray populations, and (ii) to check the soundness of the life history data that we used to parameterised the PPM. The data that we used to model growth and reproduction are sound, as several studies on the growth and reproduction of reef manta rays report very similar results (*Marshall & Bennett, 2010*; *Marshall, Dudgeon & Bennett, 2011b*; *Kashiwagi, 2014*). Less is known about individual survival in this species. *Marshall, Dudgeon & Bennett (2011b)* studied adult survival in a reef manta ray population off the southern coast of Mozambique and estimated adult survival rate at 0.68 ($\pm$0.15 SE) per year. With our own preliminary, capture-mark-recapture analysis of sight-re-sight data of adult reef manta rays off the southern coast of Mozambique (200 km south of the study site of *Marshall, Dudgeon & Bennett (2011b)*) we obtained an adult survival rate of 0.67 ($\pm$0.16 SE) per year (IBC Van der Ouderaa & Y Tibiriçá, 2014, unpublished data). From the fact that each adult survival rate estimate is within one standard error of the mean of the other estimate we infer that these two estimates are not significantly different (*Montgomery, 2012*). The survival of yearling and juvenile reef manta rays has been less studied, as individuals at these life stages do not frequently visit the aggregation sites where demographic data on adults are typically collected (*Marshall, Dudgeon & Bennett, 2011b*). For this reason, we used a range of annual survival rate values for yearlings and juveniles.

Overall, we found that the performance of our model was satisfactory; mean lifetime reproductive success and population growth rates observed in different reef manta ray populations were all within the ranges that we predicted from our population model. However, the predicted population growth rate and lifetime reproductive success values sometimes exceeded the observed values; this was probably due to the fact that we also investigated the population consequences of annual survival rates of yearlings, juveniles and adults that were lower and higher than the observed survival rates. For the reef manta ray population off the coast of the Yaeyama Islands, the annual survival rates of all three life stages, as well as the population growth rate, have been estimated: the yearling annual survival rate is estimated to be 0.63 and juvenile and adult annual survival rates are both estimated as 0.95 (*Kashiwagi, 2014*). The population growth rate associated with these values as predicted by our population model was $\sim$1.01 (Fig. 2D; $\sigma_Y = 0.63$, $\sigma_J = \sigma_A = 0.95$), which is very close to the estimated population growth rate of the Yaeyama Islands reef manta ray population of 1.02 per year (*Kashiwagi, 2014*). The only discrepancy between the predicted and observed values was cohort generation time at the lower adult annual survival rates of $0.54 \leq \sigma_A \leq 0.82$ (Figs. 4A–4C). At these low survival rates, adults do not live very

long, which lowers the average age at which adults reproduce and results in a low cohort generation time. Cohort generation time values have probably been obtained from stable populations (*Marshall et al., 2011a*; *Ward-Paige, Davis & Worm, 2013*), in which annual adult survival rates are much higher. Indeed, at $\sigma_Y = \sigma_A = 0.95$ (as found for the stable reef manta ray population off the coast of the Yaeyama Islands *Kashiwagi, 2014*), the predicted cohort generation time was 18.5 years, which is very close to the observed generation time of 19.4 years (*Ward-Paige, Davis & Worm, 2013*). Overall, it is rewarding that predictions from our population model match observations on the key population descriptors of lifetime reproductive success, population growth rate and cohort generation time.

## Demographic analyses

The demographic analysis revealed that the effects of variation in yearling and juvenile survival rates on population growth rate, mean lifetime reproductive success and cohort generation time are not straightforward, but some general patterns did emerge. Firstly, an increase in yearling or adult annual survival rate always elicited a greater response in population growth rate, mean lifetime reproductive success and cohort generation time than the same increase in juvenile annual survival rate. This suggests that a perturbation in yearling or adult annual survival rate will have far greater consequences for reef manta ray population dynamics than the same magnitude of perturbation in juvenile annual survival rate. Secondly, increases in any of the three population descriptors with increasing yearling or adult survival rate values was either linear, in the case of population growth rate, or was at an accelerating rate, in the case of mean lifetime reproductive success and cohort generation time. The accelerating rates of increase are of particular interest, because if yearling or adult annual survival rates are already high, a slight increase can increase mean lifetime reproductive success by almost one pup (Fig. 3D), and cohort generation time by a year to several years (Fig. 4D). Both of these effects can significantly affect population structure and fluctuations. Therefore, in order to obtain an accurate insight into reef manta ray population dynamics, accurate estimates of yearling and juvenile survival rates should be obtained from natural populations.

One way of gaining a general insight into the population consequences of differences in demographic rates is by using population models to project a population forward in time and investigate its future size relative to its original size, which we did for a period of 10 years for all combinations of yearling, juvenile and adult annual survival rates. The reef manta ray population off the coast of Mozambique decreased by 88% between 2005 and 2011 due to variation in the local environment, anthropogenic pressures and large-scale oceanographic influences (*Rohner et al., 2013*). Our population projections confirm that the low observed adult annual survival rate of adult reef manta rays off the coast of Mozambique of 0.68 per year (*Marshall, Dudgeon & Bennett, 2011b*) nearly always results in population decline, unless yearling and juvenile annual survival rates are close to unity. However, because reef manta ray by-catch has recently significantly increased in this region (*Marshall, Dudgeon & Bennett, 2011b*; *Pereira et al., 2014*), it is unlikely that the juvenile survival rate is close to unity. In the stable reef manta ray population off the coast of the Yaeyama Islands, the yearling annual survival rate has been estimated to be as low as 0.63,

probably because of predation (*Kashiwagi, 2014*). If we assume that this yearling annual survival rate also applies to the Mozambique population, the Mozambique population is predicted to continue to decrease in size, even if the juvenile annual survival rate is at unity (Fig. 5B). Therefore, unless the survival rates of reef manta rays in populations off the coast of Mozambique increase (by reducing direct fishing and by-catch), it is unlikely that this population will stop declining.

## Conservation implications

Many manta ray populations across the globe are declining, according to the IUCN Red List for Threatened Species (*Marshall et al., 2011a*; but see *Kashiwagi (2014)* for an exception). One way of increasing our understanding of how such declines can be reduced or even halted is by conducting elasticity analyses of a demographic model. The results of such analyses identify which demographic rates of which life stages have the greatest effect on population growth. By targeting conservation research and management on those rates and life stages, vulnerable populations can be protected from further decline (*Benton & Grant, 1999*; *Carslake, Townley & Hodgson, 2009*). Our elasticity analysis revealed that the population growth rate is most sensitive to change in either the adult survival rate or the rate at which juveniles survive but stay in the juvenile stage (i.e., do not mature into adults). To exemplify how the results of this analysis can be used, we compared the values predicted by our analysis to the values obtained in two reef manta ray populations off the coasts of Mozambique and the Yaeyama Islands. In the case of Mozambique, the observed adult annual survival rate is 0.68 (*Marshall, Dudgeon & Bennett, 2011b*), and the observed population growth rate is estimated as 0.77 per year (*Rohner et al., 2013*); according to our elasticity analysis (Fig. 2B), at these values, the population growth rate will be most sensitive to changes in the adult annual survival rate. To prevent this population declining further, the adult survival rate should be increased by reducing targeted and by-catch fishing through the protection of aggregation sites that are often frequented by adults. Our analyses indicate that if the adult survival rate increases to 0.95 per year, then the population growth rate is close to unity and the Mozambique reef manta ray population would be stable. Regarding the stable population off the coast of the Yaeyama Islands, the adult annual survival rate is 0.95, and according to our elasticity analysis, this population will also be most sensitive to changes in the adult annual survival rate. Although this population is not currently subject to direct fishing pressure (*Kashiwagi, 2014*), our results predict that any changes in adult survival will greatly affect it.

A previous demographic analysis that was based on a generic reef manta ray life cycle and not on a specific manta ray population found that the intrinsic population growth rate *r* was most sensitive to changes in the offspring production rate and not the mortality rate (*Dulvy et al., 2014*). However, unlike our elasticity analysis, *Dulvy et al.*'s (*2014*) sensitivity analysis investigated how *additive* perturbations in life history parameters affect the intrinsic population growth rate, whereas we investigated how *proportional* perturbations in demographic rates affect the long-term population growth rate. We used the second part of Eq. (4) to run a sensitivity analysis in order to investigate how *additive* perturbations affect the population growth rate, and found that the population growth rate is most

sensitive to perturbations in $G_J$ or $P_A$, depending on the values of yearling, juvenile and adult annual survival rates (Appendix S1 and Fig. S1). However, these results are not very informative, because the demographic rates in our population model are measured in different units; survival rates are probabilities, and only have values of between zero and unity, whereas reproduction rate has no such restrictions. Therefore, it is difficult to compare the sensitivity of the population growth rate to changes in survival and growth rates with the sensitivity of the reproductive rate. Therefore, we focus on the results of the elasticity analysis, which suggest that reef manta ray populations off the coast of Mozambique and Japan are most sensitive to perturbations in the adult annual survival rate. The demographic rates that comprise our population matrix are determined by the underlying parameters $\sigma_i$ (survival rate) and $\gamma_i$ (stage-specific transition rate); however, because the adult annual survival rate $P_A$ equals $\sigma_A$ and is independent of $\gamma_i$, the population growth rate is indeed most sensitive to perturbations in adult survival at high adult survival rates, which is typical for long-lived animals (*Brault & Caswell, 1993*; *Caswell, 2001*). This indicates that effective management and legislation is urgently needed to avoid the local extinction of the reef manta ray population off the coast of Mozambique. The following two approaches should be taken: (1) the species should be protected against fishing, including accidental catch; and (2) aggregation areas should be protected. The behaviour of reef manta rays at cleaning stations makes targeted fishing a potential threat, but also creates an opportunity for site-specific protection. By protecting aggregation sites, adults, which are regular visitors to such sites (*Marshall, Dudgeon & Bennett, 2011b*; *Kashiwagi, 2014*), should exhibit increased survival rates, which will result in an increase in the population growth rate.

## Alternative approaches and considerations

One aspect that we have not yet touched upon is variability in demographic rates. Such variability can, for example, occur over time through changes in environmental conditions. Exploring the population consequences of such temporal stochasticity requires in depth knowledge on how different environmental conditions affect demographic rates, of which we currently know very little. Variability in demographic rates can also manifest itself through the accuracy with which demographic rates are estimated. Such information is only available for adult annual survival rate in the Mozambique population (estimated at $\sigma_A = 0.68 \, \text{yr}^{-1} \pm 0.15$ SE) (*Marshall, Dudgeon & Bennett, 2011b*), and for non-yearling annual survival rate in the Japan population (0.95, with a 95% confidence interval of 0.94–0.96) (*Kashiwagi, 2014*). In the latter case, the accuracy of the estimated adult survival rate is very high and predictions on population dynamics as presented in panels (D) of Figs. 2–4 will vary little within the 95% confidence interval. In contrast, the accuracy of the estimated adult annual survival rate in the Mozambique population is much lower, but we can use our results to assess potential effects of this variability by exploring model output within one standard error range from the mean. That is, subtracting one standard error from the mean adult annual survival rate results in $\sigma_A = 0.68 - 0.15 = 0.53$; model output in that case will be almost equal to the results presented in panels (A) of Figs. 2–4 where $\sigma_A = 0.54$. Adding one standard error to the mean adult survival rate results

in $\sigma_A = 0.68 + 0.15 = 0.83$; model output in that case will be almost equal to the results presented in panels (C) of Figs. 2–4 where $\sigma_A = 0.82$. This comparison informs on the range of values of $\lambda$, $R_0$ and generation time that can be expected when we take the (in)accuracy of the estimated adult annual survival rate into account. One important issue is the fact that the elasticity results differ within the range $0.54 \leq \sigma_A \leq 0.82$ at intermediate values of yearling annual survival rate, $\sigma_Y$. Specifically, within the range $0.6 \leq \sigma_Y \leq 0.9$, depending on the value of $\sigma_A$, the population growth rate is either most sensitive to $P_A$, which adults survive and remain in the adult stage, or $P_J$, the rate at which juveniles survive and remain in the juvenile life stage (Fig. 2). However, if we take the estimate for yearling annual survival rate of $\sigma_Y = 0.63$ (Kashiwagi, 2014) and assume that $\sigma_A \gg 0.54$, the population growth rate is always most sensitive to perturbation of $P_A$ and our conclusions presented above on reef manta ray conservation are unaffected.

Finally, it is important to realise that different techniques exist that relate the dynamics of populations to the demographic rates of individuals, and include physiologically structured population models (Metz & Diekmann, 1986), delay-differential equation models (Nisbet & Gurney, 2003), individual-based models (Grimm & Railsback, 2005), integral projection models (IPMs) (Easterling, Ellner & Dixon, 2000), and PPMs (Caswell, 2001). These methodologies all link individual state to population structure, but differ in their mathematical approaches. Structured population models such as PPMs and IPMs are particularly useful for investigating how demographic changes affect population dynamics. They are closely and easily linked to field and experimental data, and require relatively straightforward mathematical techniques from matrix calculus (Coulson, 2012). IPMs have the added benefit that they can be used to investigate simultaneous ecological and rapid evolutionary change in quantitative characters, life history evolution and population dynamics (Smallegange & Coulson, 2013). However, IPMs are data hungry, because their parameterisation requires extensive, long-term datasets on the life history trajectories of individuals (Coulson, 2012). Because these data are currently not available for reef manta rays, we developed a PPM that included the three life stages that can currently be distinguished in reef manta rays: yearlings, juveniles and adults (Marshall et al., 2011a). Future studies should, however, aim to develop a reef manta ray IPM that can take any evolutionary responses in life history parameters to environmental change into consideration.

## ACKNOWLEDGEMENTS

We thank Hal Caswell for providing feedback on an earlier draft and Spiral Scientific Editing Services for editorial assistance.

### Funding

Isabelle B.C. van der Ouderaa was funded by the Volkert van der Willigen Foundation of the Stichting Amsterdamse Universiteits Fonds. I. M. Smallegange is funded by a MacGillavry

Fellowship from the University of Amsterdam, and a MEERVOUD grant no. 836.13.001 and VIDI grant no. 864.13.005 from the Netherlands Organisation for Scientific Research (NWO). The funders had no role in study design, data collection and analysis, decision to publish, or preparation of the manuscript.

### Grant Disclosures

The following grant information was disclosed by the authors:
Volkert van der Willigen Foundation.
MacGillavry Fellowship.
MEERVOUD: 836.13.001.
VIDI: 864.13.005.

### Competing Interests

The authors declare there are no competing interests.

### Author Contributions

- Isabel M. Smallegange analyzed the data, contributed reagents/materials/analysis tools, wrote the paper, prepared figures and/or tables, reviewed drafts of the paper.
- Isabelle B.C. van der Ouderaa analyzed the data, contributed reagents/materials/analysis tools, wrote the paper, reviewed drafts of the paper.
- Yara Tibiriçá contributed reagents/materials/analysis tools, wrote the paper, reviewed drafts of the paper.

### Data Availability

Matlab code for the demographic analyses can be accessed via Figshare: 10.6084/m9.figshare.1594759.v2.

### Supplemental Information

Supplemental information for this article can be found online at http://dx.doi.org/10.7717/peerj.2370#supplemental-information.

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
