# Peer review of "Effects of yearling, juvenile and adult survival on reef manta ray (Manta alfredi) demography"

_PeerJ, doi:10.7717/peerj.2370_

## Round 0.1 · original submission · Major Revisions

Both reviewers have issues with the methodology and validity of the findings. Please consider these carefully in a possible revision.

·

Basic reporting

The structure of the MS is generally good, although there are parts that would benefit from greater clarity and inclusion of relevant references to support assertions.

Experimental design

I n the main the design appears to be appropriate, but there is a large error in terms of one of the criteria for population modelling. I have provided comment in the annotated pdf.

The sampling period should be short in relation to the inter-sampling period, but this was not the case in this study. A sampling period of 5 months was used, which is almost the same duration as the inter-sampling period and thus the model assumption was violated.

Also, there needs be a similarity of effort, and while not explicitly presented it seems as though the effort (CPUE) varied widely between years.

The result is that the validity of the findings are very much in doubt/completely compromised. The sample size (number of resightings) is also very low for this style of analysis which raises further questions about the error associated with any population estimate.

Data/graphs of the estimated population size by year are not provided, and I expect that is because they would fluctuate hugely...due to reasons outlined above.

Validity of the findings

As mentioned above, I have grave concerns over the validity of the findings.

Marked up MS provided with more detail.

Additional comments

I would recommend that you discuss the modelling with Dr Christine Dudgeon of the University of Queensland. Also, it would be comparatively easy to determine whether the manta rays seen here are indeed the same ones as seen at Tofo that were used to model the population back in 2011. Dr Marshall should be able to provide that information.

Reviewer 2 ·

Basic reporting

The manuscript is for the most part well written although some sentences lack a clear structure and need to be rephrased for clarity (see PDF).
The background provided is relevant to the study although I found that the introduction lacked a clear flow. It feels a bit disorganised and some parts a little repetitive.
Some statements have references that are not relevant. I suggest to the authors to either review the statement so the references are appropriate to support it, or review the literature to have more relevant citations (indicated on the PDF).
I understand that PhD theses can be hard to obtain but the authors need to take into consideration the work of Kashiwagi (2014) Conservation biology and genetics of the largest living rays: manta rays PhD Thesis, School of Biomedical Sciences, The University of Queensland, that shows some key data on the population demographics of manta rays in Japan. Kashiwagi applied similar population models on data collected over a 20yrs period. This work need to be taken in consideration for the interpretations of this study

Experimental design

The research presented provides original data. The research question and research gaps are well defined, relevant and meaningful. The methods and model used are relevant to this question.
Important information is however missing. For example, the author mentioned that they only considered adult female in their models but do not provide the method used to determine maturity in females (which is notoriously hard to determine on live individuals in chondrichthyens). Some information on how laser photogrammetry was applied in the study is missing. The method is mentioned but very briefly. Did the authors only used suitable photos to apply photogrammetry? i.e photos on which the body of the manta ray is at a right angle from the laser beams’ trajectories?

Validity of the findings

It is described in the method that laser photogrammetry was used to determine the size of individuals but no data are presented in the results and there are no references to it in the discussion.
Some basic data on the population structure are also missing such as the maturity ratio observed for males (it is mentioned in the methods that this was recorded) and the size distribution of the sampled population (which will help the reader have an idea of which part of the population is mostly present at those aggregation size i.e. large individuals (adults) or small ones (juveniles)). This information is missing to support the statistical models, especially in the discussion.
The mean number of dive per year is given and is assumed equal throughout the whole study period in the results, but then in the discussion the author stated that this effort wasn’t constant among years. I think a greater attempt should be made to quantify the effort. Do the author have data on the effort made per year which would really help in interpreting the results? Information such as the number of dives done at the study sites per year and the total number of mantas seen? Or even a rough CPUE with a mean number of manta seen per dive per year? This would help standardised the data among year and give more robustness to the results.
The statistical models applied are relevant and sound.
The discussion is well stated and linked to original question but need to be reviewed in view of results from Kashiwagi 2014 and missing data identified here.

Additional comments

I have made comments directly on the PDF.

Annotated reviews are not available for download in order to protect the identity of reviewers who chose to remain anonymous.

---

## Round 0.2 · Minor Revisions

This latest review gives some issues, especially with regards to interpretation. Please consider them in a revision.

Reviewer 3 ·

Basic reporting

This is an interesting approach to investigating the effects of different survival parameters on demography of the animals. The analyses appear to be sound and the authors have used existing data. My major concerns are that the results are poorly interpreted. The authors keep stating that having different survival parameters have an affect on models but this in itself is not interesting. What these affects are and how they change dynamically are of interest and why. The limitations behind the data they use is also not discussed. There are some challenges with English grammar and wording which may be a function of second language or limited editing. these should be addressed as it makes it hard to understand the arguments.

Experimental design

This appears to be sound. The paper is largely modelling so the major data has been pulled from other published studies. The authors do not discuss any limitations of those data sources.

Validity of the findings

The results of the modelling appear to be sound. The interpretation is greatly lacking. See specific comments throughout the manuscript.

Additional comments

This is an interesting paper but is poorly executed with respect to interpreting the data. More consideration needs to be given to the actual findings of the models - how does survival affect demography and differ between different life history stages. What are the limitations of this approach and what hasn't been taken into consideration? what are the limitations of the data used from other studies? How are these findings actually significant to management recommendations? These comments need to be addressed before it is ready for publication.

Annotated reviews are not available for download in order to protect the identity of reviewers who chose to remain anonymous.

---

## Round 0.3 · Minor Revisions

Along with one of the previous reviewers we had an additional reviewer look carefully at the analyses. Both have come back with some remaining minor comments for you to consider.

Reviewer 3 ·

Basic reporting

see below

Experimental design

see below

Validity of the findings

see below

Additional comments

This is a revision of a manuscript that I previously reviewed. I am happy with the changes that the authors have made in addressing previous issues. the manuscript has been improved substantially and I think it is ready for publication following minor revision and independent assessment of statistical modelling.



Lines 50 – 53: The first sentence is very stilted. Given the paper is about manta rays and not seahorses etc. these should be given as examples and not listed as such.
Line 60: re-word. Manta rays have ‘a slow life history’. Not a generally used term. They have life history strategies that result in…..
Line 92: this sentence contradicts the previous one about adult survival being important in turtles and killer whales. The previous sentence does not say anything about juveniles – only adults. Be consistent.
Line 116: variation
Line 276: variation
Line 362: given you do not present any statistical tests between your survival rate and that from Marshall et al., do not sure the term significant here. Just say that your survival estimate is very similar to that one from Marshall et al.
Lines 357-363: very long sentence. Break up.
Line 379: remove the term ‘actual’ population growth rate. This is also only an estimate so state that your results are similar to the estimates generated from Kashiwagi.
Line 394: here and throughout the manuscript – variation is plural as well as singular. Change from variations
Lines 432-456: in both of these elasticity analyses, the adult survival rate is emphasised as being the most important. This doesn’t seem to be reflected in the summary of the paper.
Line 421: Presumably Dulvy et al. had to run their sensitivity analyses based on similar types of data – how did they overcome the problems with rates being in different units?
Line 481: why at the national level? Is this based on an assumption that this level covers population level processes? Then that should be specified. Depending on the nation in question there could be more than one population that needs to be protected.
Table 2: Tropical Eastern Pacific

·

Basic reporting

The preparation of the manuscript is at a high standard. The English reads well and the ideas and concepts are presented clearly. Information presented is sufficient to understand the context of the problem and how the objectives of the investigation addresses the problem.

Experimental design

The investigation is a modelling exercise and based on published literature. There appear to be no problems with the sources of information or how they were used to build population matrix models. Methods are clear and understandable. There is sufficient detail to allow for replication.

Validity of the findings

I take some issue with the approach being used, and offer an alternative that I think better accounts for uncertainty and variability in a poorly-understood natural system.

It appears the authors treated parameter values in their PPM in a deterministic manner (i.e. assuming that values for survival, transition, reproduction remain fixed through each analysis/forward simulation). What would happen if stochasticity were incorporated into the analysis?

I see stochasticity as the remaining variability in a data set that has not been explained by variables that researchers measure. Incorporating left-over variability explicitly seems particularly important for poorly-understood systems with much unexplained variation, such as those that are the topic of this paper.

There are two important sources that should be considered in a modelling exercise, particularly when it is being used to make decisions about species conservation. One relates to variability in the process (i.e. that demographic rates vary over time in response to environmental factors). The other relates to the accuracy with which parameters are estimated (SEs or 95% CIs). Models that don't at least attempt to incorporate this variability will tend to produce over-optimistic (i.e. too precise) output.

For example, assuming one pup every two years is reasonable for a large population, but as populations decrease, the proportion of females that breed in any year, and how this varies from year to year, can have an important effect on subsequent dynamics. How might the findings differ if F_A = 0.5 were replaced with a binomial distribution, where p = 0.5 and N = the number of reproductive females in the simulated population?

All of the estimates are associated with standard errors that provide a picture of certainty about the parameter estimate (sampling error), but they also describe a distribution within which demographic estimates might vary. Demographic rates might be assumed to be constant over the period of a projection, but what happens if they are allowed to vary according to a distribution, having a mean and variance set by parameters in the literature?

If the same simulation with parameters-as-distributions is run repeatedly, what proportion of runs suggest an increase or a decrease? This is different that changing the values of survival and fecundity and then leaving them at fixed values (which is important too). What if those values were treated as means and represented with normal distributions?

How might the certainty of your results change if lambda were treated as a distribution of with a range of potential values that could arise from survival and fecundity estimates that incorporate variability? What if the elasticities are treated as distributions rather than a single fixed outcome to set levels of the demographic inputs?

I'm not familiar with Matlab, but I imagine the individual fixed parameter values in eqns 1—3, 5, 6 could be replaced by distributions having particular parameter values? This would be fairly simple in R.

Additional comments

The only additional comment has to do with the beginning of the Discussion where the authors compare their approach to other modelling approaches. The comparison is useful, but I would place it later in the Discussion, perhaps right before the conservation implications. What I generally look for at the beginning of the Discussion is a summary or synthesis of the main findings, as a reminder before the authors go into the explanation of their findings and comparisons with the literature. As it is, this is a little out of place.

---

## Round 0.4 · accepted · Accept

Thank you for your corrections and explanation as to why you did some things as you did.